# CAR-Based Immunotherapy of Solid Tumours—A Survey of the Emerging Targets

**DOI:** 10.3390/cancers15041171

**Published:** 2023-02-11

**Authors:** John Maher, David M. Davies

**Affiliations:** 1CAR Mechanics Group, Guy’s Cancer Centre, School of Cancer and Pharmaceutical Sciences, King’s College London, Great Maze Pond, London SE1 9RT, UK; 2Department of Immunology, Eastbourne Hospital, Kings Drive, Eastbourne BN21 2UD, UK; 3Leucid Bio Ltd., Guy’s Hospital, Great Maze Pond, London SE1 9RT, UK

**Keywords:** solid tumour, chimeric antigen receptor, T-cell, NK cell, target, toxicity

## Abstract

**Simple Summary:**

A rapidly emerging new approach to treat cancer involves collection of patient immune white blood cells and genetic re-programming of the cells to express a new cancer-detecting receptor called a CAR. This approach has revolutionized the treatment of some blood cancers, whereas solid tumours, which account for 90% of all cancers, are proving much more difficult to treat using this approach. A major challenge in this respect relates to the identification of targets that differentiate cancer from healthy cells. In a partner review, we consider clinical data already collected when CAR technology has been applied against solid tumours that express 30 different targets. Here, we explore emerging candidates for which such clinical data are not available yet, but where other data provide information about potential suitability as future clinical targets.

**Abstract:**

Immunotherapy with CAR T-cells has revolutionised the treatment of B-cell and plasma cell-derived cancers. However, solid tumours present a much greater challenge for treatment using CAR-engineered immune cells. In a partner review, we have surveyed data generated in clinical trials in which patients with solid tumours that expressed any of 30 discrete targets were treated with CAR-based immunotherapy. That exercise confirms that efficacy of this approach falls well behind that seen in haematological malignancies, while significant toxic events have also been reported. Here, we consider approximately 60 additional candidates for which such clinical data are not available yet, but where pre-clinical data have provided support for their advancement to clinical evaluation as CAR target antigens.

## 1. Introduction

Chimeric antigen receptors are fusion proteins that comprise a targeting moiety (most commonly a single chain antibody fragment (scFv)) followed by a hinge/spacer, a transmembrane domain and a signalling unit. Using this approach, specificity of immune cell such as T-cells can be re-programmed against a desired tumour-associated cell surface target antigen. Second generation CARs are defined by an endodomain in which a single co-stimulatory unit (most commonly from CD28 or 4-1BB) is placed upstream of an ITAM (immune tyrosine activation motif)-containing activating module. Immunotherapy using second generation CAR T-cells has achieved unprecedented impact in patients with relapsed refractory B-cell and plasma cell malignancies [1,2]. However, solid tumours account for about 90% of human cancers and largely remain impervious to these approaches. The first of many challenges in this context is target selection, in particular, since truly tumour-specific antigens are extremely difficult to source. For this reason, investigators generally evaluate tumour-associated antigens. In a partner review article, we have surveyed published clinical experience of CAR T-cell immunotherapy directed at 30 target antigens that are found on solid tumours (Table 1) [3]. However, several additional targets have been evaluated in pre-clinical and ongoing unpublished studies of CAR T-cell immunotherapy. It should be noted that none of these targets are absolutely tumour specific and thus all constitute tumour-associated antigens. Here, we have considered these emerging candidate targets with respect to expression in normal tissue and solid tumours, together with pre-clinical/ongoing clinical CAR evaluation (Table 2) and clinical data when related immunotherapeutic interventions were employed against these targets. We have focused exclusively on cell surface targets, while excluding HLA-restricted CAR T-cell approaches, given the need, in this case, for appropriate HLA matching between patient and CAR. It should be noted that CARs may also be targeted against HLA-restricted neoantigens [4], and the interested reader is also referred to [5] for an overview of HLA-restricted CAR-based immunotherapeutic approaches.

## 2. Receptor Tyrosine Kinases

### 2.1. Fibroblast Growth Factor Receptor 4 (FGFR4)

Fibroblast growth factor receptor 4 (FGFR4) is primarily expressed during embryonic muscle development, and Protein Atlas confirms that it is found at low levels in multiple adult tissues (https://www.proteinatlas.org/ENSG00000160867-FGFR4/tissue, accessed on 3 January 2023) [6]. By contrast, FGFR4 is over-expressed in basal-like breast, liver, colon, head and neck and prostate cancers, in addition to both fusion positive and fusion negative rhabdomyosarcoma. In rhabdomyosarcoma, FGFR4 promotes both metastasis and therapeutic resistance. Chimeric antigen receptors targeted against FGFR4 have been described by a number of groups [7,8]. Potential safety of this target is further supported by clinical evaluation of an FGFR4 inhibitor, which is well tolerated [9]. No active CAR clinical trials directed against FGFR4 are registered at clinicaltrials.gov (accessed on 18 December 2022).

### 2.2. Erythropoietin-Producing Hepatocellular Carcinoma (Ephrin) Receptors

Ephrin receptors represent the largest group of receptor tyrosine kinases, numbering 14 in total, which are classified into A and B subgroups. They interact with Ephrin ligands on neighbouring cells, thereby exerting their physiological roles in tissue organization and development. Ephrin dysregulation has been linked not only to malignant, but also non-malignant disease states. For example, several Ephrins are found in atherosclerotic lesions and during tissue fibrosis [10]. Moreover, expression of many of these receptors in healthy tissues has been reported at the mRNA level [11]. EphA2 is already the subject of clinical evaluation using CAR T-cell immunotherapy [12], and a number of other Ephrins have been evaluated in pre-clinical CAR T-cell research, as summarized below. No active CAR clinical trials directed against the Ephrin receptors listed below were identified on clinicaltrials.gov (accessed on 21 December 2022).

#### 2.2.1. Ephrin Type A3 Receptor (EphA3)

EphA3 is intimately involved in a number of developmental processes [10]. It is also highly expressed in neovasculature and stromal elements found in glioblastomas, and it is over-expressed and/or mutated in many other solid tumours, generally associated with worsened prognosis [10]. By contrast, EphA3 expression in adult tissues is generally low level (https://www.proteinatlas.org/ENSG00000160867-FGFR4/tissue, accessed on 3 January 2023), with highest expression in the uterus, bladder and prostate [11]. CAR T-cells directed against this target have been generated using an scFv derived from the EphA3-specific antibody, ifabotuzumab [13]. A Phase 1 trial with this antibody in patients with haematological malignancies demonstrated good safety and clinical activity, with a number of responses [14]. In glioblastoma patients, zirconium 89-labelled ifabotuzumab demonstrated excellent tumour localization with little normal tissue uptake, as determined by positron emission tomographic (PET) imaging [15].

#### 2.2.2. Ephrin Type A10 Receptor (EphA10)

EphA10 is reported to be expressed at a low level in normal adult tissue, except the testis and skin [16] (https://www.proteinatlas.org/ENSG00000183317-EPHA10/tissue#expression_summary, accessed on 3 January 2023). Aberrant expression of EphA10 has been described in a number of solid tumours including luminal breast cancer [17], prostate, lung and ovarian tumours, and it is also found on tumour-associated macrophages and myeloid-derived suppressor cells. Moreover, EphA10 expression has been linked to poorer prognosis. A panel of antibodies has recently been generated against this target, one of which has been used to engineer CAR T-cells that demonstrated anti-tumour activity in models of triple negative breast cancer (TNBC) [16].

#### 2.2.3. Ephrin Type B4 Receptor (EphB4)

EphB4 is expressed on a variety of tumours including rhabdomyosarcoma [18]. By contrast, it is reported to be expressed at low to moderate levels in some normal tissues [19], most notably the gastrointestinal tract (https://www.proteinatlas.org/ENSG00000196411-EPHB4/tissue, accessed on 3 January 2023). A CAR directed against this target has recently been generated using the natural ligand Ephrin B2 to direct target specificity [18,19]. Cells demonstrated pre-clinical anti-tumour activity [18] and were safe over a 7-day period following infusion into lymphodepleted non-human primates [19]. Soluble EphB4-albumin blocks bidirectional signalling between EphB4 and Ephrin B2 and was well tolerated when administered to patients with head and neck cancer [20]. 

### 2.3. Receptor Tyrosine Kinase-Like Orphan Receptor Family Member (ROR)2

Receptor tyrosine kinase-like orphan receptors (ROR) are receptors for non-canonical Wnt ligands. They are expressed during embryonic development and at low levels in adult tissue (no protein expression data available on Protein Atlas, accessed on 3 January 2023), although they are upregulated during inflammation and fibrosis [21]. ROR1 is now being evaluated as a clinical CAR T-cell target, while ROR2 remains the subject of pre-clinical studies. ROR2 expression has been described in non-small cell lung cancer (NSCLC), melanoma, squamous cell carcinoma of head and neck (SCCHN), osteosarcoma, neuroblastoma and renal cell carcinoma [22], and CAR T-cells with anti-tumour activity have been generated against this target [23]. An interim analysis of a Phase 2 antibody drug conjugate targeting ROR2 in patients with NSCLC is expected in 2023 (https://biopharmaapac.com/news/31/2177/himalaya-therapeutics-announces-highlights-of-recent-clinical-progress.html, accessed on 21 December 2022).

### 2.4. ALK Receptor Tyrosine Kinase

The ALK receptor is implicated in a number of developmental processes and is expressed in most neuroblastomas in addition to rhabdomyosarcoma, Ewing’s sarcoma, melanoma, NSCLC, glioblastoma, prostate cancer, breast cancer and anaplastic large cell lymphoma. It is reported to be virtually absent in normal tissues, although Protein Atlas reports expression in brain and skin (https://www.proteinatlas.org/ENSG00000171094-ALK/tissue, accessed on 21 December 2022). ALK acts as an oncogenic driver and is linked to worsened prognosis, providing a selective pressure for maintained expression. Pre-clinical studies have successfully demonstrated that ALK can be targeted using CAR T-cell immunotherapy, although cell surface levels may be too low in some neuroblastomas for productive engagement by some CAR-targeting moieties [24]. ALK inhibitors have been used in the treatment of neuroblastoma, and they lead to target upregulation, potentially further increasing efficacy of CAR T-cell targeting. Moreover, tumours with low ALK expression may also be targeted effectively using logic AND gate-based CAR T-cells in which ALK binding was coupled to delivery of an activating signal by a first generation CAR [25]. This was co-expressed with a chimeric co-stimulatory receptor in which CD28 signalling was coupled to the binding of either B7-H3 or GD2. No actively recruiting CAR clinical trials targeting ALK were listed on clinicatrials.gov (accessed on 21 December 2022).

### 2.5. AXL Receptor Tyrosine Kinase

The AXL receptor is expressed in several cancer types, including chronic myeloid leukaemia (CML), acute myeloid leukaemia (AML), breast cancer, NSCLC, renal cell carcinoma (RCC), thyroid cancer, liver cancer, oesophageal cancer and pancreatic ductal adenocarcinoma (PDAC) [26]. It is pro-tumourigenic, promotes epithelial to mesenchymal transition and is generally associated with poorer prognosis. Despite functions in neurogenesis and immune cells, monoclonal antibodies and small molecule inhibitors of this receptor are generally well tolerated in man and have achieved anti-tumour activity [26]. Nonetheless, Protein Atlas highlights AXL expression in several organs and that it is found in immune cell types, including immunosuppressive macrophage and myeloid cell populations [27,28] (https://www.proteinatlas.org/ENSG00000167601-AXL/tissue, accessed on 22 December 2022). Pre-clinical studies have demonstrated anti-tumour activity of AXL-targeted CAR T-cells, either alone [29,30] or in combination with tumour-targeted microwave ablation [26]. Clinical trials of AXL-targeted CAR T-cells known to be currently recruiting are listed in Table 3.

### 2.6. Platelet-Derived Growth Factor Receptor (PDGFR) α

Platelet-derived growth factor receptor (PDGFR) α is expressed in paediatric brain tumours and sarcomas such as rhabdomyosarcoma [31,32]. Protein Atlas indicates expression in the normal gastrointestinal tract and bone marrow/lymphoid tissue (https://www.proteinatlas.org/ENSG00000134853-PDGFRA/tissue, accessed on 3 January 2023). Nonetheless, the PDGFR α-specific monoclonal antibody Olaratumab proved to be well tolerated in man. However, this agent has been withdrawn from the market for sarcoma treatment owing to failure to enhance patient survival when combined with doxorubicin chemotherapy [33]. CAR T-cells with pre-clinical anti-tumour activity against PDGFR have recently been described [34]. No active CAR clinical trials directed against this target were identified on clinicatrials.gov (accessed on 22 December 2022).

### 2.7. Protein Tyrosine Kinase 7 (PTK7; Colon Carcinoma Kinase 4; CCK4)

PTK7 is a Wnt family pseudokinase that is enriched on cancer stem cells in a range of solid tumours, including TNBC, NSCLC, gastrointestinal tumours and ovarian cancer. Expression is associated with worsened clinical outcome. Normal cells that express PTK7 include some immune cell subsets, with modest expression in other healthy tissues (https://www.proteinatlas.org/ENSG00000112655-PTK7/tissue, accessed on 22 December 2022), most notably the stomach [35]. PTK7 has been targeted using an auristatin-based antibody–drug conjugate [36]. The safety profile was acceptable and clinical responses were seen in about 20% of patients, depending on tumour type. CAR T-cells directed against PTK7 have recently been described [35]. No active CAR clinical trials directed against this target were identified on clinicatrials.gov (accessed on 22 December 2022).

## 3. Adhesion Molecules

### 3.1. Intracellular Adhesion Molecule 1 (ICAM-1)

Intracellular adhesion molecule 1 (ICAM-1) is upregulated in several solid tumours, although it is also found in many normal tissues, most notably the lung, kidney and endometrium (https://www.proteinatlas.org/ENSG00000090339-ICAM1/tissue, accessed on 7 January 2023). To discriminate between upregulated expression in tumours and background healthy tissue expression, low-affinity CARs with micromolar k_d_ have been designed [37]. This CAR is currently undergoing Phase 1 evaluation in patients with relapsed refractory thyroid cancer (NCT04420754, accessed on 7 January 2023). No other actively recruiting trials of ICAM-1-specific CAR T-cell immunotherapy were identified on clinicatrials.gov (accessed on 7 January 2023). An ICAM-1-specific monoclonal antibody was well tolerated in Phase 2 testing in patients with smouldering multiple myeloma, although no clinical responses were achieved in this patient population [38].

### 3.2. Cadherins

Cadherins are primarily expressed on the basal aspect of epidermal cells, where they mediate calcium-dependent adhesion.

#### 3.2.1. Cadherin 6

Cadherin 6 is also known as foetal kidney cadherin and is primarily involved in renal development, while expression in normal adult tissues is mainly restricted to some renal tubules and hepatic bile ducts, with lower-level expression in other tissues [39] (https://www.proteinatlas.org/ENSG00000113361-CDH6/tissue, accessed on 7 January 2023). Expression is upregulated in renal cell, ovarian and thyroid cancers. One pre-clinical study has described the development of CAR T-cells targeted against Cadherin 6 [39]. No active CAR clinical trials directed against this target were identified on clinicatrials.gov (accessed on 7 January 2023). Phase 1 testing of a Cadherin 6-specific antibody–drug conjugate was terminated prematurely because of unexplained neurotoxicity observed during dose escalation [40].

#### 3.2.2. Cadherin 17

Cadherin 17 is upregulated in hepatic, gastrointestinal and pancreatic cancers in addition to neuroendocrine tumours [41]. Although cadherin 17 is expressed at high levels by normal intestinal cells (https://www.proteinatlas.org/ENSG00000079112-CDH17/tissue, accessed on 7 January 2023), access is restricted since it is only localized at inaccessible tight junctions. Consequently, CAR T-cells targeted against cadherin 17 mediate anti-tumour activity without toxicity in pre-clinical models [41]. This CAR is under development by Chimeric Therapeutics (https://chimerictherapeutics.com/our-pipeline/#our-pipeline, accessed on 7 January 2023). No additional active CAR clinical trials directed against this target were identified on clinicatrials.gov (accessed on 7 January 2023).

### 3.3. Nectin-4 (Poliovirus Receptor-like 4)

Nectin-4 (Poliovirus receptor-like 4) is an adhesion molecule that is highly expressed in bladder, breast, pancreatic, ovarian, lung, renal cell, melanoma and head and neck cancers [42]. Strongest staining is reported in urothelial and breast cancers. It exerts a number of pro-tumourigenic effects, and expression is associated with adverse prognosis in some studies [42]. Expression is restricted to embryonic development and Nectin-4 is found at low levels in adult tissues including normal oesophagus (https://www.proteinatlas.org/ENSG00000143217-NECTIN4/tissue, accessed on 10 January 2023). CAR T-cells targeted against Nectin-4 have recently been described [43]. Moreover, an antibody–drug conjugate has been approved in multiple territories for the treatment of locally advanced/metastatic bladder cancer [44]. However, recent data suggest that Nectin-4 expression declines in urothelial malignancy following metastatic spread [45]. No active CAR clinical trials directed against this target were identified on clinicatrials.gov (accessed on 10 January 2023).

### 3.4. CD44 v6 Splice Variant (CD44v6)

CD44 is an adhesion receptor that binds hyaluronic acid. The v6 splice variant that includes exon 6 binds hepatocyte growth factor and has been implicated in disease progression in a range of malignancies including AML, multiple myeloma and solid tumours such as pancreatic, breast, colon, ovarian, lung and head and neck cancers [46]. Owing to its ability to bind HGF, it promotes MET activation, in addition to modulating the activity of other receptor types [47]. CD44v6 is absent in haematopoietic stem cells, but is expressed at low levels in activated T-cells, monocytes, keratinocytes and oral mucosa [48] (expression data unavailable on Protein Atlas, accessed on 10 January 2023). There have been several pre-clinical reports describing CD44v6-targeted CAR T-cells [48,49,50,51]. Evaluation of CD44v6 as a clinical target has encountered mixed results. An antibody–drug conjugate, bivatuzumab, was evaluated in patients with SCCHN or oesophageal carcinoma [52,53]. However, skin toxicity was identified, manifesting as toxic epidermal necrolysis in one fatal case, prompting discontinuation of drug development. Clinical trials of CD44v6-targeted CAR T-cells known to be currently recruiting are listed in Table 4. 

## 4. Integrins

Integrins are heterodimeric adhesion molecules that are involved in migration and invasion by tumour cells in addition to angiogenesis.

### 4.1. αvβ6 Integrin

αvβ6 integrin is an epithelial-specific integrin that is aberrantly expressed in a wide range of epithelial tumours, including PDAC and squamous cell carcinomas of various organs. It is absent from healthy adult epithelial tissues, although expression is upregulated during wound healing. Protein Atlas suggests that this integrin is expressed at low levels in adult tissue with the exception of the bladder, where higher-level expression is seen (https://www.proteinatlas.org/ENSG00000115221-ITGB6/tissue, accessed on 3 January 2023). It is linked to worsened prognosis at least in part owing to its role in the activation of latent transforming growth factor (TGF)-β, which is a potently immunosuppressive cytokine. CAR T-cells have been engineered to target this integrin alone [54,55,56,57,58], or armoured with the CXCR2 chemokine receptor [59]. Interim data from a Phase 1 clinical trial of an antibody–drug conjugate targeted against this integrin have demonstrated acceptable safety thus far, with emerging evidence of efficacy [60]. No active CAR clinical trials directed against this target were identified on clinicatrials.gov (accessed on 5 January 2023).

### 4.2. αvβ3 Integrin

αvβ3 integrin is typically found on newly formed endothelial cells (including tumour-associated neovasculature), but it is also expressed by a number of tumour types, particularly those that occur in the central nervous system, melanoma, breast, pancreatic and prostate cancer. Small molecules and antibodies directed against αvβ3 integrin did not show anti-tumour activity, but, nonetheless, proved safe in man. Protein Atlas indicates that the main site of physiological expression of this integrin is in bone marrow (https://www.proteinatlas.org/ENSG00000259207-ITGB3/tissue, accessed on 3 January 2022). αvβ3 integrin-targeted CAR T-cells have recently been described and exhibited anti-tumour activity in pre-clinical models of intracranial [61] and extracranial tumours [62,63]. No active CAR clinical trials directed against this target were identified on clinicatrials.gov (accessed on 5 January 2023).

## 5. Vascular Targets

### 5.1. CLEC14A

CLEC14A is a shear-induced endothelial marker that exhibits low-level expression in adult tissues, mainly in the endothelium and pulmonary alveoli [64]. Nonetheless, Protein Atlas highlights expression in a number of sites, most notably skin, gastrointestinal and genitourinary tissues (https://www.proteinatlas.org/ENSG00000176435-CLEC14A/tissue, accessed on 5 January 2023). While poorly expressed under conditions of physiological blood flow, it is upregulated in several solid tumours. To exploit this, CAR T-cells have recently been generated against this target [65]. Despite cross-reactive specificity for both human and mouse CLEC14A, CAR T-cells were non-toxic in mice and exhibited anti-tumour activity in three distinct models [65]. No active CAR clinical trials directed against this target were identified on clinicatrials.gov (accessed on 5 January 2023).

### 5.2. Apelin Receptor

The Apelin receptor is expressed in tumour-associated neovasculature [66]. By contrast, normal tissue expression is limited and mainly cytoplasmic in nature (https://www.proteinatlas.org/ENSG00000134817-APLNR/tissue, accessed on 5 January 2023). Synthetic (Syn)Notch CAR T-cells have been engineered, which couple the recognition of Apelin engagement to transcription of a second CAR with tumour specificity [67]. No active CAR clinical trials directed against this target were identified on clinicatrials.gov (accessed on 5 January 2023).

## 6. Extracellular Matrix Targets

### 6.1. Fibronectin with Extra Domain B (EDB)

Fibronectin with extra domain B (EDB) is an oncofoetal fibronectin that is linked to tissue regeneration and angiogenesis. By contrast, EDB is rarely found in healthy tissues [68]. Expression of EDB in the tumour microenvironment is induced by TGF-β [69]. It is found in the extracellular matrix of most solid tumours, lymphomas and some leukaemias [70,71] and is secreted by tumour cells, cancer-associated fibroblasts and neovascular endothelial cells. Since it adheres to the cell surface in an RGD-dependent manner, it thereby “paints” the tumour microenvironment. This is particularly the case for subendothelial extracellular matrices of newly formed blood vessels, although tumour cells stain much less intensively if at all [72]. Production of EDB potentiates endothelin-1 expression in endothelial cells, which, in turn, triggers a positive feedback loop whereby EDB released by malignant and endothelial cells is further boosted. Expression of EDB has been linked with poorer prognosis in human cancer [73]. However, expression within tumours may be heterogeneous, with the most intense expression in the periphery rather than centre of lesions [74]. Expression data in Protein Atlas for this target were not available (accessed on 5 January 2023).

CAR T-cells have been generated to target EDB using the L19 antibody and have been shown to recognise both tumour cells and tumour-associated vasculature, achieving pre-clinical anti-tumour efficacy [75,76]. Although human umbilical vein endothelial cells (HUVEC) are also destroyed by EDB-specific CAR T-cells, mouse studies support the safety of this target, which has an identical protein sequence in both mouse and man [75,76]. Similarly, an EDB-specific antibody–drug conjugate was well tolerated in cynomolgous monkeys at doses of up to 12 mg/kg, when haematological and corneal toxicity emerged [71]. No active CAR clinical trials directed against this target were identified on clinicatrials.gov (accessed on 5 January 2023).

### 6.2. Fibronectin with Extra Domain A (EDA)

Fibronectin with extra domain A (EDA) is also an oncofoetal fibronectin that is rarely found in healthy adult tissues, with the exception of the endometrium and ovary [77]. Expression data in Protein Atlas for this target were not available (accessed on 6 January 2023). EDA expression is upregulated in a broad range of tumours in which prognosis is adversely affected [78]. Similarly to EDB, the human and mouse protein sequence is identical. No toxicity has been seen when EDA-specific CAR T-cells were infused in mice, while anti-tumour efficacy was apparent [79]. No active CAR clinical trials directed against this target were identified on clinicatrials.gov (accessed on 5 January 2023).

## 7. Gangliosides

Clinical experience of CAR T-cell targeting of disialoganglioside (GD2) is summarized elsewhere [3]. Here, we consider alternative tumour-associated gangliosides for CAR targeting. 

### 7.1. Ganglioside D3

Ganglioside D3 is expressed during CNS development and is aberrantly expressed in a range of tumours, including melanoma (over 80%), neuroblastoma, glioma, and cancers of the cervix, lung, prostate, breast, head and neck, colon and ovary [80]. Low-level expression of GD3 has been described on retinal pigment cells, in the CNS and on normal melanocytes [81]. Expression data in Protein Atlas for this target were not available (accessed on 6 January 2023). GD3 is pro-tumourigenic and leads to accelerated tumour growth [82]. Accordingly, GD3-specific CAR T-cells have been evaluated in a number of pre-clinical studies with evidence of anti-tumour activity in the absence of toxicity [80,83]. No active CAR clinical trials directed against this target were identified on clinicatrials.gov (accessed on 6 January 2023).

### 7.2. Ganglioside M2

Ganglioside M2 (GM2) is expressed by several solid tumours including melanoma, neuroblastoma, myeloma, sarcoma and renal carcinoma. Expression data in Protein Atlas for this target were not available (accessed on 6 January 2023). Noile-Immune Biotech are undertaking a Phase 1 clinical trial of “Proliferation-Inducing and Migration-Enhancing” (PRIME) CAR T-cells (armoured with a combination of IL-7 and CCL19) in patients with GM2-expressing solid tumours (https://clinicaltrials.gov/ct2/show/NCT05192174, accessed on 6 January 2023). Use of IL-7 and CCL19 armouring promotes accumulation of the CAR T-cells in solid tumours [84]. 

## 8. B7 Family Members

### 8.1. B7-H4

B7-H4 is highly expressed in a number of solid tumours, including approximately 50% of breast, ovarian and endometrial cancers [85]. However, expression in normal tissue has been the subject of varied reports [86] and data were not available in Protein Atlas (accessed on 6 January 2023). Pre-clinical testing of B7-H4 CAR T-cells in models of ovarian cancer demonstrated anti-tumour activity [86]. However, delayed toxicity was evident in treated mice, accompanied by multiorgan CAR T-cell infiltration and clear evidence of B7-H4 expression in many tissues. Analysis of human tissues in this study confirmed that B7-H4 was widely expressed in normal organs. By contrast, Phase 1 testing of a B7-H4 specific monoclonal antibody did not result in any dose-limiting toxicities [85]. Clinical trials of B7-H4-CD3 bispecific antibodies (NCT05067972; NCT05180474) and a B7-H4-specific antibody–drug conjugate (NCT05263479) are ongoing. No active CAR clinical trials directed against this target were identified on clinicatrials.gov (accessed on 7 January 2023).

### 8.2. B7-H6

B7-H6 is recognized by the NKp30 receptor, as are BAT3/ BAG6 and galectin 3—all of which are commonly expressed on tumour cells [87]. Data on protein expression were not available on Protein Atlas, although mRNA was detected in many tissues (data pending verification; https://www.proteinatlas.org/ENSG00000188211-NCR3LG1/tissue, accessed on 7 January 2023). Accordingly, NKp30 has been employed to direct CAR T-cell specificity against B7-H6 in pre-clinical studies [88]. A first in man clinical trial of a B7-H6/CD3 T-cell engager is currently underway [89]. No active CAR clinical trials directed against this target were identified on clinicatrials.gov (accessed on 7 January 2023).

## 9. Prostate Antigens

### 9.1. Kallikrein-Related Peptidase 2 (KLK2)

Kallikrein-related peptidase 2 (KLK2; also known as human kallikrein 2) is expressed by normal prostate tissue, but is not expressed at other sites (https://www.proteinatlas.org/ENSG00000167751-KLK2/tissue, accessed on 7 January 2023). Expression is further upregulated in prostate cancer. Johnson & Johnson are developing a suite of prostate cancer therapies directed against this target, including a bispecific antibody, radiotherapy antibody–drug conjugate and CAR T-cells derived from induced pluripotent stem cells (iPSC) (https://www.fiercepharma.com/pharma/johnson-johnson-plays-up-precision-therapy-for-zejula-s-half-successful-prostate-cancer-bid, accessed on 16 January 2023). The latter have demonstrated pre-clinical anti-tumour activity [90] and are currently undergoing clinical evaluation (NCT05022849). No additional active CAR clinical trials directed against this target were identified on clinicatrials.gov (accessed on 7 January 2023).

### 9.2. Six-Transmembrane Epithelial Antigen of Prostate (STEAP) Family Members

Six-transmembrane epithelial antigen of prostate (STEAP) 1–4 consist of a family of five metalloproteinases, namely STEAP1, 1B, 2, 3 and 4, which (with the exception of STEAP3) are commonly expressed in prostate cancer [91].

#### 9.2.1. Six-Transmembrane Epithelial Antigen of Prostate (STEAP) 1

Six-transmembrane epithelial antigen of prostate-1 (STEAP1) is over-expressed in approximately 90% of prostate cancers [92]. It is also found in a subset of other cancer types in which it exerts pro-tumourigenic actions [91]. In normal tissue, it is mainly expressed in the prostate gland. However, Protein Atlas also reports significant protein expression in the respiratory and central nervous system (https://www.proteinatlas.org/ENSG00000164647-STEAP1/tissue, accessed on 8 January 2023). Targeting of STEAP1 with an antibody–drug conjugate was well tolerated, with no evidence of target-dependent toxicity. Evidence of efficacy in some patients was indicated by reduction in circulating prostate-specific antigen level [91]. A T-cell engager directed against this target is currently undergoing Phase 1 evaluation [91] (NCT04221542; sponsored by Amgen). In addition, STEAP1-specific CAR T-cells with pre-clinical anti-tumour activity have recently been described [92]. No active CAR clinical trials directed against this target were identified on clinicatrials.gov (accessed on 8 January 2023).

#### 9.2.2. Six-Transmembrane Epithelial Antigen of Prostate (STEAP) 2

Six-transmembrane epithelial antigen of prostate-2 (STEAP2) is also over-expressed in prostate cancer and, similar to STEAP1, it is also pro-tumourigenic [91]. It is also expressed in other tumours including osteosarcoma. Astra Zeneca are believed to be developing a CAR T-cell program directed against this target (AZD0754), while Regeneron have patent filings pertaining to an STEAP2 antibody and derived therapeutics (https://www.evaluate.com/vantage/articles/news/corporate-strategy/pulling-back-curtain-astrazenecas-car-t-work, accessed on 8 January 2023). Protein Atlas reports widespread low-level expression of STEAP2 in many tissues (https://www.proteinatlas.org/ENSG00000157214-STEAP2/tissue, accessed on 8 January 2023). No active CAR clinical trials directed against this target were identified on clinicatrials.gov (accessed on 8 January 2023).

## 10. Notch Ligands

The Notch pathway plays a key role in many developmental processes, acting through juxtacrine signalling between neighbouring cells. Notch signalling is commonly dysfunctional in human cancer. Two Notch ligands have been the subjects of pre-clinical CAR engineering.

### 10.1. Delta-like Ligand 3 (DLL3)

Delta-like ligand 3 (DLL3) is an inhibitory Notch ligand and is upregulated on neuroendocrine tumours such as small cell lung cancer [93]. Expression in normal tissues is generally low and mainly cytoplasmic [93] (protein expression data unavailable on Protein Atlas, accessed on 8 January 2023). Targeting of this antigen has a somewhat checkered history. The antibody–drug conjugate rovalpituzumab tesirine (Rova)-T failed in Phase 3 testing in patients with small cell lung cancer, since it did not achieve either a safety or efficacy benefit [94]. Interim results of a DLL3-targeted trispecific T-cell engager in patients with neuroendocrine tumours indicated that it was well tolerated, although only 1 partial response was seen in 16 enrolled patients [95]. The Amgen bispecific T-cell engager AMG 757 (tarlatamab) achieved a response rate of 13% in 64 treated patients (63% at the highest dose level), although toxicity was notable, with 21 toxic events at grade 3 or above, including 1 death due to pneumonitis [96]. An Amgen-sponsored Phase 1 study of AMG 119 CAR T-cells targeted against DLL3 in small cell lung cancer was suspended with no patients enrolled (NCT03392064; posted November 2022, accessed on 8 January 2023). There is also an ongoing clinical trial of CAR NK cells targeted against DLL3 in patients with small cell lung cancer (NCT05507593, accessed on 8 January 2023). No additional active CAR clinical trials directed against this target were identified on clinicatrials.gov (accessed on 8 January 2023). 

### 10.2. Delta-like Homologue 1 (DLK1)

Delta-like homologue 1 is highly expressed in hepatocellular carcinoma, neuroblastoma and tumours arising from the lung, breast, brain, pancreas and colon [97]. Pre-clinical studies of CAR T-cells directed against this target confirmed anti-tumour activity [98]. Regarding normal tissue expression, it is primarily found in the adrenal glands (https://www.proteinatlas.org/ENSG00000185559-DLK1/tissue, accessed on 8 January 2023). No active CAR clinical trials directed against this target were identified on clinicatrials.gov (accessed on 8 January 2023).

## 11. Viral Antigens

### 11.1. Human Endogenous Retrovirus (HERV) K

Human endogenous retrovirus accounts for approximately 8% of human genomic DNA [99]. Expression is strictly controlled in normal tissues, but not in some disease states, and there have been multiple reports indicating that HERV K proteins are expressed in a number of tumour types [100]. However, recent data indicate that HERV-K gene expression also occurs in a range of normal tissues [101]. CAR T-cells targeted against HERV-K have been reported [102,103]. No active CAR clinical trials directed against this target were identified on clinicatrials.gov (accessed on 8 January 2023).

### 11.2. Latent Membrane Protein 1 (LMP1)

Latent membrane protein 1 (LMP1) is an Epstein Barr virus (EBV)-encoded cell surface protein and is commonly expressed in EBV-related malignancies such as nasopharyngeal carcinoma. LMP-specific CARs have been described, which demonstrated anti-tumour activity in models of EBV-related cancer [104,105,106,107]. No active CAR clinical trials directed against this target were identified on clinicatrials.gov (accessed on 8 January 2023).

## 12. Glypicans

Glypicans are oncofoetal heparan sulphate proteoglycans that are involved in a number of developmental processes and are aberrantly expressed in a range of tumours. Glypican 3 is already the subject of a number of CAR T-cell clinical trials, as summarised elsewhere [3]. Here, we consider remaining family members for which pre-clinical CAR data only have been generated. 

### 12.1. Glypican 1

Glypican 1 is over-expressed in pancreatic, oesophageal, prostate, breast, cervical and ovarian cancers in addition to glioma and malignant mesothelioma [108]. It is associated with a number of pro-tumourigenic actions. A CAR with specificity for both human and mouse glypican 1 has been described and elicited anti-tumour activity both in vitro and in vivo [109]. Protein Atlas reports that glypican 1 is expressed in the skin (and at lower levels in other tissues; https://www.proteinatlas.org/ENSG00000063660-GPC1/tissue, accessed on 16 January 2023), although CAR T-cells that recognize the mouse orthologoue of this target did not elicit any evidence of toxicity [109]. No active CAR clinical trials directed against this target were identified on clinicatrials.gov (accessed on 8 January 2023).

### 12.2. Glypican 2

Glypican 2 is expressed in neuroblastoma, paediatric brain tumours and other childhood cancers [110,111]. Normal tissue expression is reported to be restricted to lower levels in the skin and oesophagus [110]. Expression data in Protein Atlas were not available (https://www.proteinatlas.org/ENSG00000213420-GPC2/tissue, accessed on 8 January 2023). CAR T-cells with specificity for Glypican 2 have been engineered with the ability to detect even low levels of this antigen without induction of on-target off-tumour toxicity in mice, despite the potential to bind to the mouse orthologue of this target [110,111]. No active CAR clinical trials directed against this target were identified on clinicatrials.gov (accessed on 8 January 2023). 

## 13. Placental Antigens

### 13.1. Placental Alkaline Phosphatase (PLAP)

Placental alkaline phosphatase (PLAP) is a placental tumour antigen. It is expressed minimally in normal tissues, primarily in reproductive organs [112]. Protein Atlas also highlights strong expression in the normal gastrointestinal tract (https://www.proteinatlas.org/ENSG00000163283-ALPP/tissue, accessed on 11 January 2023), although this is refuted in part by mRNA expression data in [113]. By contrast, it is aberrantly expressed in germ cell, cervical, colorectal, endometrial, ovarian and gastric tumours [112]. Chimeric antigen receptors with PLAP specificity and anti-tumour activity have been described [112,113]. No active CAR clinical trials directed against this target were identified on clinicatrials.gov (accessed on 11 January 2023).

### 13.2. Trophoblast Cell Surface Antigen 2 (TROP2)

Trophoblast cell surface antigen 2 (TROP2) is a transmembrane calcium signal transducer and is expressed in most solid tumour types including breast, cervical, colorectal, pancreatic, prostate, lung, head and neck and gastric cancer. Physiologically, it is expressed in trophoblast, prostate stem cells and facultative stem cells known as liver oval cells [114]. Expression data were not available on Protein Atlas (accessed on 11 January 2023). Administration of an antibody–drug conjugate targeted against TROP2 was well tolerated and achieved a 19% response rate in patients with non-small cell lung cancer [115]. This agent, known as Sacituzumab govitecan-hziy (Trodelvy^®^, Gilead, Foster City, CA, USA), has been approved for the third line treatment of TNBC based on an overall response rate of 33% and acceptable tolerability profile [116,117]. It is also approved for treatment of bladder/urothelial cancer following platinum chemotherapy and PD(L)1 blockade. A second TROP2-specific antibody–drug conjugate, datopotamab deruxtecan, has also achieved a 34% response rate and acceptable safety in metastatic TNBC patients [117]. However, interstitial lung disease was noted in NSCLC patients treated with this drug, including three grade 5 toxic events, although this may be related to the payload rather than the target [118]. CAR T-cells with TROP2 specificity have been developed and demonstrated anti-tumour activity in pre-clinical testing [114,119,120]. No active CAR clinical trials directed against this target were identified on clinicatrials.gov (accessed on 11 January 2023).

### 13.3. Human Leucocyte Antigen G (HLA-G)

Human leucocyte antigen G (HLA-G) is a non-classical HLA class I antigen and is expressed on trophoblast cells and by a range of tumour types, including renal cell, ovarian, colorectal, lung, endometrial and pancreatic cancers [121,122]. It acts as an immune checkpoint by engaging ILT2, ILT4 and KIR2DL4 immune inhibitory receptors, in addition to CD8, which it blocks [122]. HLA-G is naturally expressed at minimal levels in adult tissues (https://www.proteinatlas.org/ENSG00000204632-HLA-G/tissue, accessed on 11 January 2023), although expression in the pituitary gland has been noted [123]. CAR T-cells targeted against HLA-G have recently been evaluated pre-clinically [121]. A clinical trial of a bispecific HLA-G/CD3 antibody is currently ongoing [123]. Although there are no actively recruiting CAR clinical trials, one study has recently been posted on clinicaltrials.gov in which HLA-G-specific CAR T-cells will be administered to patients with solid tumours that express this target (NCT05672459; sponsored by MD Anderson Cancer Center in collaboration with Invectys).

### 13.4. 5T4 (Trophoblast Glycoprotein)

5T4 (trophoblast glycoprotein) is highly expressed by trophoblast in addition to a wide range of solid tumours (e.g., renal, lung, breast, ovarian, endometrial, bladder, PDAC, oesophageal, gastric and colorectal carcinomas in addition to malignant mesothelioma). Protein Atlas reports significant expression in many tissues, although cytoplasmic in many cases (https://www.proteinatlas.org/ENSG00000146242-TPBG/tissue, accessed on 12 January 2023). This target is associated with cancer stem cell properties and, in some cancers, stromal expression has also been reported [124]. CAR T-cells with 5T4-dependent anti-tumour activity have been described by a number of groups [125,126]. A Phase 1 trial of a 5T4-specific antibody–drug conjugate did not reveal toxicities attributable to the target itself [127]. Clinical trials of 5T4-targeted CARs known to be currently recruiting are listed in Table 5.

## 14. Lewis Y

Lewis Y is a carbohydrate antigen that is overexpressed on a range of solid tumours including colon, breast, prostate, lung and ovarian carcinomas, as well as AML. It is also found at lower levels in normal epithelial surfaces, with a polarized distribution that makes it relatively inaccessible to CAR T-cells [128]. A small clinical trial in AML demonstrated the safety of Lewis Y CAR T-cells [129]. Some transient responses were noted, followed by rapid progression. A trial of Lewis Y-targeted CAR T-cells in solid tumour patients has been completed, but outcomes remain unpublished (NCT03851146). A clinical trial in which a humanized anti-Lewis Y antibody was administered to patients with breast cancer was terminated owing to insufficient efficacy [130]. One actively recruiting CAR T-cell trial was identified in which Lewis Y is one of multiple targets across a range of solid tumours (NCT03198052).

## 15. MG7

MG7 is a gastrointestinal tumour antigen against which a CAR has been described in an abstract publication [131]. A clinical trial of MG7-specific CAR T-cells in patients with liver metastases is listed as of unknown status on clinicaltrials.gov (NCT02862704). A further abstract may pertain to this product, since it describes the combined intratumoural and intravenous administration of MG7-specific CAR T-cells in a patient with metastatic colorectal cancer [132]. No actively recruiting CAR clinical trials directed against this target were identified on clinicatrials.gov (accessed on 9 January 2023).

## 16. Glial Cell Line-Derived Neurotrophic Factor Family Receptor α4 (GFR α4)

Glial cell line-derived neurotrophic factor family receptor α4 (GFR α4) exhibits highly restricted expression to normal parafollicular cells in the thyroid and derived medullary carcinomas, which are the third most common form of thyroid cancer [133]. To exploit this, a GFR α4-specific CAR has been engineered [133]. A clinical trial of this approach is currently ongoing at the University of Pennsylvania (NCT04877613). No other active CAR clinical trials directed against this target were identified on clinicatrials.gov (accessed on 9 January 2023).

## 17. Leucine-Rich Repeat-Containing G Protein-Coupled Receptor 5 (LGR5)

Leucine-rich repeat-containing G protein-coupled receptor 5 (LGR5) is an orphan receptor and a stem cell marker [134]. Expression is upregulated in a number of solid tumours, including hepatocellular, ovarian, colorectal and breast carcinomas and glioblastoma multiforme. Protein Atlas shows expression of LGR5 in the respiratory tract and, at lower levels, in other tissues (https://www.proteinatlas.org/ENSG00000139292-LGR5/tissue, accessed on 9 January 2023). A Phase 1 trial of an anti-LGR5 monoclonal antibody in colorectal cancer was terminated by the sponsor (NCT02726334; Bionomics Ltd. Adelaide, Australia). Pre-clinical anti-tumour activity of LGR5-specific CAR T-cells has recently been demonstrated in models of ovarian cancer [135], while LGR5-specific CAR NK cells have also been described [136]. No active CAR clinical trials directed against this target were identified on clinicatrials.gov (accessed on 9 January 2023), although Carina Biotech Pty Ltd. have received a safe to proceed notification for an Investigational New Drug application to the US FDA for Phase 1/2a testing of LGR5 CAR T-cells in patients with colorectal cancer (https://www.yahoo.com/now/carina-biotech-receives-fda-safe-235500803.html, accessed on 24 January 2023). This CAR has been generated using the humanized monoclonal antibody earlier developed and tested by Bionomics Ltd. 

## 18. Non-Functioning P2X7 Receptor (nfP2X7)

Non-functioning P2X7 receptor (nfP2X7) is a conformational variant that is expressed on a broad range of tumour types, owing to exposure of a normally hidden epitope [137,138]. Epitope exposure protects tumour cells from the high ambient concentration of ATP in the tumour microenvironment. Weak membrane staining was observed in less than 10% of normal tissues, whereas 40% of tumours showed continuous expression [139]. No expression data for this target were available on Protein Atlas (accessed on 9 January 2023). An nfP27X-specific CAR is being developed by Biosceptre Ltd. Cambridge, UK and is currently in lead optimization (https://www.biosceptre.com/pipeline/, accessed on 9 January 2023). No other active CAR clinical trials directed against this target were identified on clinicaltrials.gov (accessed on 9 January 2023).

## 19. Follicle-Stimulating Hormone Receptor (FSHR)

Follicle-stimulating hormone receptor (FSHR) is expressed on up to 70% of ovarian cancers of various subtypes, but is absent on non-ovarian healthy tissue [140] (RNA data only available at Protein Atlas; https://www.proteinatlas.org/ENSG00000170820-FSHR/tissue, accessed on 9 January 2023). To exploit this, CAR T-cells have been targeted against FSHR using binding fragments derived from the ligand FSH [141]. A Phase 1 clinical trial of this approach is currently ongoing in patients with ovarian cancer (NCT05316129). 

## 20. Guanylyl Cyclase 2C (GUCY2C)

Guanylyl cyclase 2C (GUCY2C) is expressed in a polarized manner on the apical brush border of intestinal epithelial cells, where it is relatively inaccessible to circulating CAR T-cells (https://www.proteinatlas.org/ENSG00000070019-GUCY2C/tissue, accessed on 9 January 2023). Expression of GUCY2C is upregulated and lacking such polarity in virtually all colorectal carcinomas in addition to other gastrointestinal and pancreatic malignancies. It is also expressed on a subset of hypothalamic neurons. Pre-clinical studies have demonstrated the therapeutic potential and in vivo safety of GUCY2C-targeted CAR T-cells [142,143]. A Phase 1 clinical trial of a GUCY2C antibody–drug conjugate highlighted risk of hepatotoxicity, one episode of which was fatal [144]. Two actively recruiting CAR T-cell clinical trials were identified in patients with gastrointestinal tumours (NCT05287165; NCT04652219).

## 21. Thyroid-Stimulating Hormone Receptor (TSHR)

Thyroid-stimulating hormone receptor (TSHR) is highly and homogeneously expressed on the majority of thyroid carcinomas, with negligible expression in other tissues (https://www.proteinatlas.org/ENSG00000165409-TSHR/tissue, accessed on 10 January 2023). Successful pre-clinical targeting of TSHR using CAR T-cells has recently been described [145]. No active CAR clinical trials directed against this target were identified on clinicatrials.gov (accessed on 9 January 2023).

## 22. C-Type Lectin 4 Family (CLEC4)

Members of the C-type lectin 4 family (CLEC4) are upregulated in hepatocellular carcinoma [146]. A patent filing to protect CLEC4-specific CARs has been filed by Atara Biotherapeutics, Thousand Oaks, CA, USA (WO2020227595A1; https://patentimages.storage.googleapis.com/32/4a/41/9b74ec2949eaa3/WO2020227595A1.pdf, accessed on 10 January 2023).

## 23. CD47

CD47 is widely expressed on multiple tumour types and imparts a “don’t eat me” signal to signal regulatory protein (SIRP) α-expressing phagocytic cells [147]. However, it is expressed at low levels by many normal cells including T-cells (https://www.proteinatlas.org/ENSG00000196776-CD47/tissue, accessed on 11 January 2023). CAR T-cells with specificity for CD47 have been developed and characterised in a number of pre-clinical studies [148,149,150]. In some cases, fratricide due to recognition of CD47 on T-cells resulted in poorer yields of CAR T-cells, in a manner that was abrogated if the CAR endodomain was truncated [149]. Phase 1 testing of an anti-CD47 monoclonal antibody was generally well tolerated, despite temporary holds placed on a number of studies by the Food and Drug Administration. Anaemia was the most predominant adverse reaction [151,152]. No active CAR clinical trials directed against this target were identified on clinicatrials.gov (accessed on 11 January 2023).

## 24. Transmembrane 4 L6 Family Member 1 (TM4SF1)

Transmembrane 4 L6 family member 1 (TM4SF1) is a tetraspannin that is highly expressed in a range of epithelial cancers including pancreatic, liver, lung, oesophageal breast and bladder cancers [153]. It is reputed to maintain cancer stemness and to promote epithelial to mesenchymal transition and angiogenesis [153]. Phase 1 testing of a monoclonal antibody with specificity for TM4SF1 did not reveal any significant toxicity concerns [153]. The protein is expressed at modest levels in a number of normal tissues (https://www.proteinatlas.org/ENSG00000169908-TM4SF1/tissue, accessed on 11 January 2023). A clinical trial of TM4SF1-specific CAR T-cells is listed as of unknown status (NCT04151186). 

## 25. A Disintegrin and Metalloproteinase 10 (ADAM10)

A disintegrin and metalloproteinase 10 is expressed at moderate levels in a number of normal tissues (https://www.proteinatlas.org/ENSG00000137845-ADAM10/tissue, accessed on 12 January 2023). However, a conformational variant of ADAM10 has been reported in a number of tumours, including colorectal cancer [154]. Efficacy of CAR T-cells directed against this target has recently been reported in abstract form [154]. No active CAR clinical trials directed against this target were identified on clinicatrials.gov (accessed on 12 January 2023).

## 26. Chlorotoxin Ligands

Chlorotoxin is a 36 amino acid peptide derived from the venom of the death stalker scorpion. It binds to multiple membrane proteins on tumours such as glioblastoma, while showing negligible binding to normal cell types [155]. Matrix metalloproteinase 2 is believed to be the primary mediator of chlorotoxin binding. To exploit this, CAR T-cells have been engineered in which specificity of targeting is directed by chlorotoxin [155]. One actively recruiting clinical trial of this approach is ongoing at the City of Hope Cancer Center, Duarte, CA, USA in which intraventricular or intracavitary delivery of CAR T-cells is performed in patients with malignant glioma/glioblastoma (NCT04214392). Dosing at the third level was reported last year in this study (https://chimerictherapeutics.com/wp-content/uploads/2022/02/3RD-DOSE-LEVEL-INITIATED-IN-CLTX-CAR-T-PHASE-1-TRIAL-.pdf, accessed on 12 January 2023).

## 27. Heat Shock Protein 70 (HSP70)

Heat shock protein 70 (HSP70) is a stress-induced protein that exhibits strict tumour-specific cell surface expression, owing to differential expression of certain lipids in the plasma membrane of transformed cells. Cell surface expression has been described in several tumours including colorectal and gastric carcinomas and various squamous cell carcinomas, while normal tissue was negative for cell surface HSP70 in this study [156]. CAR T-cells targeted against membrane-anchored HSP70 have been described [157]. No active CAR clinical trials directed against this target were identified on clinicatrials.gov (accessed on 12 January 2023).

## 28. Cripto-1

Cripto-1 is a GPI-anchored foetal oncoprotein that plays an important developmental role, is a marker of cancer stem cells and has been detected in several common solid tumour types [158]. By contrast, expression in normal adult tissues is reported to be low [159]. Protein expression data were not available at Protein Atlas (accessed on 12 January 2023). Monoclonal antibodies targeted against Cripto-1 are available [159] with a view to generating derived therapeutics, including CARs [159]. No active CAR clinical trials directed against this target were identified on clinicatrials.gov (accessed on 12 January 2023).

## 29. Glucose-Regulated Protein 78 (GRP78)

Glucose-regulated protein 78 (GRP78) is an endoplasmic reticulum chaperone protein that is involved in the unfolded protein response. Protein Atlas confirms widespread expression in normal tissues with a cytoplasmic distribution (https://www.proteinatlas.org/ENSG00000044574-HSPA5/tissue, accessed on 14 January 2023). It is upregulated on the cell surface in several solid tumours and AML [160]. It also associates with Cripto-1. GRP78-specific CARs have been developed by two groups and demonstrated therapeutic activity in models of AML, while sparing normal haematopoietic stem cells [161,162]. Anti-tumour function was impressively potentiated by expansion of the CAR T-cells in dasatinib [161]. Phase 1 testing of a GRP78-specific monoclonal antibody was well tolerated in patients with multiple myeloma [163]. No active CAR clinical trials directed against this target were identified on clinicatrials.gov (accessed on 14 January 2023).

## 30. CD147 (Basigin)

CD147 (Basigin) is highly expressed in a range of cancers including NSCLC, breast cancer, hepatocellular carcinoma and T-cell acute lymphoblastic leukaemia. It exerts a number of pro-tumourigenic actions including enhanced proliferation, invasiveness and metastatic potential [164]. However, it is expressed at lower levels in a range of normal tissues (https://www.proteinatlas.org/ENSG00000172270-BSG/tissue, accessed on 14 January 2023). Nonetheless, Phase 1/2 testing of a CD147-specific monoclonal antibody revealed a good safety profile and evidence of efficacy in patients with COVID-19 infection, exploiting the role of CD147 as a receptor for this virus [165]. CAR T-cells targeted against CD147 have achieved anti-tumour activity in pre-clinical models of hepatocellular carcinoma, employing doxycycline inducibility [166] or SynNotch technology [167] to mitigate on-target off-tumour toxicity. CD147-specific CAR T-cells have also shown anti-tumour activity in models of NSCLC [168]. A single CAR T-cell clinical trial is listed on clinicaltrials.gov, targeting CD147 in T-cell non-Hodgkin’s lymphoma, although recruitment has not yet opened (NCT05013372). 

## 31. CD317 (Tetherin)

CD317 (Tetherin) is expressed by a broad range of tumours and confers resistance to apoptosis [169]. However, the protein is expressed in a range of normal tissues, including various immune cell types (https://www.proteinatlas.org/ENSG00000130303-BST2/tissue, accessed on 14 January 2023). A CAR with specificity for this target has been developed and demonstrated anti-tumour activity in models of glioblastoma [170]. No active CAR clinical trials directed against this target were identified on clinicatrials.gov (accessed on 14 January 2023).

## 32. Chondroitin Sulphate Proteoglycan 4 (CSPG4)

Chondroitin sulphate proteoglycan 4 (CSPG4) is highly expressed with limited heterogeneity in several solid tumours, including melanoma (over 80% of cases; [171]), leukaemias, glioblastoma, TNBC, head and neck cancer, mesothelioma and sarcomas [172,173]. It is expressed both on cancer stem cells and also tumour vasculature (pericytes) in some cases [172]. However, Protein Atlas indicates that it is also widely expressed in many normal tissues (https://www.proteinatlas.org/ENSG00000173546-CSPG4/tissue, accessed on 14 January 2023). In some, but not all cancers, it exerts oncogenic driver activities [173]. Chimeric antigen receptors targeted against CSPG4 have been described by multiple groups [171,172,173,174]. No active CAR clinical trials directed against this target were identified on clinicatrials.gov (accessed on 14 January 2023).

## 33. CD24

CD24 is a cancer stem cell marker and is also expressed by B- and T-cells, monocytes, granulocytes, epithelial, neural and muscle cells [175] (https://www.proteinatlas.org/ENSG00000272398-CD24/tissue, accessed on 15 January 2023). It has been linked with perturbed signalling in transformed cells and is upregulated in several tumour types, including breast, lung, liver, colorectal, pancreatic, ovarian, bladder, prostate, brain and head and neck carcinomas [175]. Anti-tumour efficacy of CD24-specific CARs has been demonstrated in both T-cells [176] and NK cells [177]. When evaluated clinically, a CD24-specific monoclonal antibody was well tolerated [175]. No active CAR clinical trials directed against this target were identified on clinicatrials.gov (accessed on 14 January 2023).

## 34. Müllerian-Inhibiting Substance Type 2 Receptor (MISIIR)

Müllerian-inhibiting substance type 2 receptor (MISIIR) is a member of the TGF-β receptor family and plays a key role in the regression of the primordial female reproductive apparatus in the male embryo. MISIIR is expressed by the majority of ovarian and endometrial carcinomas as well as a number of non-gynaecological solid tumour types [178]. Expression in normal tissues is primarily restricted to granulosa (ovary), Sertoli and Leydig (tests) cells, with very low-level expression reported in other tissues [178] (protein expression data unavailable on Protein Atlas, accessed on 15 January 2023). A CAR with anti-tumour activity directed against this target has recently been described [178]. Moreover, MISIIR re-directed CAR T-cells were non-toxic when cultured with a panel of healthy human cell types or when infused into mouse xenograft models, despite the ability of the CAR to recognize the mouse ortholog of this target antigen. No active CAR clinical trials directed against this target were identified on clinicatrials.gov (accessed on 15 January 2023).

## 35. SLC3A2 (Solute Carrier Family 3 Member 2)/CD98hc (Heavy Chain)

SLC3A2 (solute carrier family 3 member 2)/CD98hc (heavy chain) is the target antigen recognized by the SF-25 antibody, which binds to colorectal tumour cells but not normal counterparts [179]. It is a component of heteromeric amino acid transporters, which enable cells to meet nutritional and anti-oxidant requirements via the enhanced uptake of branched chain and aromatic amino acids [180]. Expression of SLC3A2 is also found in breast and genitourinary cancers and malignant melanoma. Normal tissue expression is restricted primarily to the kidney and testis, although Protein Atlas also highlights its presence in a number of other tissues (https://www.proteinatlas.org/ENSG00000168003-SLC3A2/tissue, accessed on 15 January 2023). CAR T-cells directed against this target exhibited anti-tumour activity both in vitro and in vivo [179]. No active CAR clinical trials directed against this target were identified on clinicatrials.gov (accessed on 15 January 2023).

## 36. Conclusions

Cell therapy now constitutes the largest area of drug development within the immune-oncology arena [181]. Within this setting, CAR T-cell immunotherapy is the largest sector with the greatest year on year growth in clinical trial activity [181]. Nonetheless, efficacy against solid tumours has been unsatisfactory, and tumour-specific target antigens remain elusive. This is reflected in the fact that 30 targets have been explored in solid tumour-focused clinical trials of CAR T-cell immunotherapy [3], while approximately 60 additional targets have been the subjects of pre-clinical interest. A key further challenge is the need for CAR T-cells to traffic to and extravasate into tumour deposits, which they need to infiltrate effectively while withstanding the metabolically and immunologically hostile tumour microenvironment [182].

Many of the emerging targets presented here are expressed at low levels in healthy tissues and may represent promising clinical targets in specific solid tumour indications. Based on clinical trial data presented to date, most promising results have been reported with targets that exhibit relatively more tumour-specific expression, such as claudin 18.2 [183]. Tumour specificity of expression is likely to be a key factor in success, since this should increase safe dosing levels in addition to minimising CAR T-cell stimulation in normal tissues, thereby reducing exhaustion prior to tumour encounter. In this context, several potentially promising onco-foetal targets are presented in this review. Insight into potential safety of these candidates is also informed by clinical data obtained using other biologics.

While targeting of some individual solid tumour targets has achieved impressive success [183], it seems unlikely that there is another “CD19-type” target remaining to be discovered in the broad solid tumour area. Instead, different forms of innovation are likely to be required to deliver meaningful success in this quest. More sophisticated targeting strategies that couple sustained immune effector potency to the recognition of two or more targets and armouring technologies that re-program the tumour microenvironment and help to stimulate endogenous anti-tumour immunity represent some of the approaches that may help to address this logjam. In this context, selection of the right target may be more about providing the therapeutic cells with a “first approximation” of what differentiates tumour deposits from healthy tissue, allowing additional engineered components of the therapeutic cell product to deepen therapeutic impact and selectivity in the quest for a cure. 

## Figures and Tables

**Table 1 cancers-15-01171-t001:** Solid tumour CAR targets that have been evaluated clinically.

Target Class	Target
Glycosylphosphatidyl-inositol-anchored cell surface protein	Mesothelin
	Carcinoembryonic antigen
Receptor tyrosine kinase	HER2
	Epidermal growth factor receptor (EGFR)
	EGFR variant III
	Receptor tyrosine kinase-like orphan receptor family member (ROR) 1
	c-Met
	Vascular endothelial growth factor receptor (VEGFR) 2
	Erythropoietin-producing human hepatocellular carcinoma type A receptor 2 (EphA2)
Mucins	Mucin (MUC) 1
	MUC 16
	Tumour-associated glycoprotein (TAG) 72
Claudins	Claudin 6
	Claudin 18.2
Folate receptors	Folate receptor α
Interleukin (IL) receptor	IL-13 receptor α2
Gangliosides	GD2
B7 family members	B7-H3
	PD-L1
Glypicans	Glypican 3
Prostate antigens	Prostate-specific membrane antigen
	Prostate stem cell antigen
Adhesion molecules	Epithelial cell adhesion molecule (EpCAM)
	Neuronal L1 cell adhesion molecule (L1CAM)
Miscellaneous	NKG2D ligands
	CD70
	Carboxy anhydrase IX
	Fibroblast activation protein
	CD133
	Roundabout guidance receptor (ROBO) 1

**Table 2 cancers-15-01171-t002:** Non-clinical and ongoing clinical evaluation of emerging solid tumour CAR targets.

Target	Pre-Clinical In Vitro	Pre-Clinical In Vivo	Ongoing Clinical Trial
FGFR4	X	X	
EphA3	X		
EphA10	X	X	
EphB4	X	X	
ROR2	X		
ALK	X	X	
AXL	X	X	X
PDGFR	X	X	
PTK7	X	X	
ICAM-1	X	X	
Cadherin 6	X		
Cadherin 17	X	X	
Nectin-4	X	X	
CD44v6	X	X	X
αvβ6 integrin	X	X	
αvβ3 integrin	X	X	
CLEC14A	X	X	
Apelin receptor	X	X	
EDB	X	X	
EDA	X	X	
GD3	X	X	
GM2	X	X	X
B7-H4	X	X	
B7-H6	X	X	
KLK2	X	X	
STEAP1	X	X	
STEAP2	Not reported	Not reported	
DLL3	Not reported	Not reported	X
DLK1	X	X	
HERV K	X	X	
LMP1	X	X	
Glypican 1	X	X	
Glypican 2	X	X	
TROP2	X	X	
HLA-G	X	X	X
5T4	X	X	X
Lewis Y	X	X	X
MG7	Not reported	Not reported	X
GFR α4	X	X	X
LGR5	X	X	In preparation
nfP2X7	Not reported	Not reported	
FSHR	X	X	X
GUCY2C	X	X	X
TSHR	X	X	
CLEC4	Not reported	Not reported	
CD47	X	X	
TM4SF1	Not reported	Not reported	X
ADAM10	X	X	
Chlorotoxin ligands	X	X	X
HSP70	X		
Cripto-1	Not reported	Not reported	
GRP78	X	X	
CD147	X	X	X (non-Hodgkin’s lymphoma)
CD317	X	X	
CSPG4	X	X	
CD24	X	X	
MISIIR	X	X	
SLC3A2	X	X	

**Table 3 cancers-15-01171-t003:** Ongoing CAR T-cell clinical trials directed against AXL (accessed on 3 January 2023).

Disease	Sponsor	Notes	Identifier
AXL + solid tumours	2nd Affiliated Hospital Guangzhou Medical University	One of multiple targets	NCT03198052
AXL + sarcomas	Shanghai PerHum Therapeutics	CXCR5 armoured CAR T-cells	NCT05128786

**Table 4 cancers-15-01171-t004:** Ongoing CAR T-cell clinical trials directed against CD44v6 (accessed on 10 January 2023).

Disease	Sponsor	Notes	Identifier
CD44v6 + Solid tumours	Shenzhen Geno-Immune Medical Institute		NCT04427449
CD44v6 + Breast cancer	Shenzhen Geno-Immune Medical Institute	One of multiple targets	NCT04430595

**Table 5 cancers-15-01171-t005:** Ongoing CAR T-cell clinical trials directed against 5T4 (accessed on 12 January 2023).

Disease	Sponsor	Notes	Identifier
Solid tumours	Shanghai East Hospital	Allogeneic NK host cells	NCT05137275
Solid tumours	Wuxi People’s Hospital	NK host cells	NCT05194709

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
