# Peer review of "CAR-Based Immunotherapy of Solid Tumours—A Survey of the Emerging Targets"

_cancers, 2023, doi:10.3390/cancers15041171_

Round 1

Reviewer 1 Report

The authors extensively reviewed potential targets  for CAR-based immunotherapy for solid tumors. Most targets in this article have not been investigated in clinical trials. Over 60 targets have been chosen in their review and discussion. The article is well written and the material objectively presented. Indeed, such a summary is a good reference for investigators in this field. I have several comments. 

1. The article was addressed by plain description of all potential targets. From a reader's point of view, the authors should emphasize some important issues. For example, are some targets more important than others? Which target will most likely be clinically relevant? 

2. Clinical trials are ongoing for at least two targets (AXL and CD44v6 ). The cancer types should be described. 

3. There are different phases of pre-clinical studies. For some targets, the potential appears to be based in speculation (purely theoretical). In others, there are evidence shown by in vitro studies. In better cases, animal studies have been done. I think the level of pre-clinical evidence can be shown explicitly (for example, by a summary table). 

Minor

The legend of table 2 may be incorrect (pasted from legend of table 1?)

Author Response

We thank reviewer 1 for their comments.

  1. We have extended the conclusions section to highlight targets that we feel are most likely to achieve promising clinical results. text now reads: "Based on clinical trial data presented to date, most promising results have been reported with targets that exhibit relatively more tumour-specific expression, such as claudin 18.2 [180]. Tumour specificity of expression is likely to be a key factor in success since this should increase safe dosing levels in addition to minimising CAR T-cell stimulation in normal tissues tissues, thereby reducing exhaustion prior to tumour encounter. In this context, several potentially promising onco-foetal targets are presented in this review.
  2. Table 1 has been amended to indicate that AXL+ solid tumours and AXL+ sarcomas are being treated with AXL-specific CAR T-cells. Similarly, Table 2 has been amended to indicate that CD44v6+ solid tours and CD44v6+ breast cancers are being treated with CD44v6-specific CAR T-cells.
  3. A summary Table has been added as requested.
  4. The error in the legend has been corrected. 

Reviewer 2 Report

This review gives an overview of potential CAR target antigens of solid tumours. Target selection is important since solid tumours are difficult to treat with CAR- based immunotherapy.

The manuscript is well-written and - organised.

Author Response

Thanks you for your comments.

Reviewer 3 Report

Comments for Maher and Davies

Maher and Davies in the review titled “CAR based immunotherapy of solid tumors – survey of the emerging targets” discussed the current progress with CAR-T therapy in solid tumors pertaining to the class of targetable antigens available for therapeutic interventions.

Major and minor comments are noted below:

Major points:

1.       the review needs to discuss more in detail the challenges related to the use of tumor associated antigens (TAAs) vs tumor specific antigens (TSAs) for CAR-T therapy. The use of TAA is used due to the lack of TSAs for CAR-T therapy in solid tumors.

2.       Could the concept of neoantigens for CAR-T therapy in solid tumors be discussed in this review?

3.       The conclusion paragraph would be more comprehensive if the inherent bottleneck attributed to the tumor microenvironment (TME). This affects the recognition, trafficking, and survival of the CAR-T cells. More importantly, CAR-T cells may not be able to penetrate tumor tissue through the vascular endothelium.

Minor points:

1.       Would be helpful to incorporate a table incorporating the different types of CAR-T therapy for solid tumors, and their associated clinical trials as discussed in the review.

2.       Please explain the significance of mentioning “in a partner review article, we have surveyed published clinical experience of CAR T-cell immunotherapy directed at 30 target antigens that are found on solid tumors (Mahar & Davis, under review” if the article is yet to be published when readers are not able to find out that those 30 target antigens are. It would be beneficial at least to mention that types/class of these 30 target antigens belongs.

Author Response

We thanks reviewer 3 for their helpful comments.   1. To address this point, we have expanded the Introduction which now reads:  "The first of many challenges in this context is target selection, in particular since truly tumour-specific antigens are extremely difficult to source. For this reason, investigators generally evaluate tumour-associated antigens. In a partner review article, we have surveyed published clinical experience of CAR T-cell immunotherapy directed at 30 target antigens that are found on solid tumours (summarized in Table 1; Maher & Davies, under review). However, several additional targets have been evaluated in pre-clinical and ongoing unpublished studies of CAR T-cell immunotherapy. It should be noted that none of these targets are absolutely tumour specific and this all constitute tumour-associated antigens. Here, we have considered these emerging candidate targets with respect to expression in normal tissue and solid tumours, together with pre-clinical/ ongoing clinical CAR evaluation (Table 2) and clinical data when related immunotherapeutic interventions were employed against these targets."   2. We have briefly touched on CARs targeted against Neo-antigens although these are generally HLA-restricted and thus beyond the defined scope of the review. Text has been amended to read: "We have focused exclusively on cell surface targets while excluding HLA-restricted CAR T-cell approaches, given the need in this case for appropriate HLA matching between patient and CAR. It should be noted that CARs may also be targeted against HLA-restricted neoantigens [3] and the interested reader is also referred to [4] for an overview of HLA-restricted CAR-based immunotherapeutic approaches."   3. To address this point, the conclusion has been modified to incorporate the following text: "A key further challenge is the need for CAR T-cells to traffic to and extravasate into tumour deposits, which they need to infiltrate effectively while withstanding the metabolically and immunologically hostile tumour microenvironment [181]."   Minor 1-2. Tables have been added to list existing clinical stage targets (including their class) and also which emerging targets are the subject of ongoing clinical trials.